



**Exploring the inconsistent variations in atmospheric primary and**
**secondary pollutants during the G20 2016 Summit in Hangzhou, China:**
**implications from observation and model**
**Gen Zhang[1*], Honghui Xu[2*], Hongli Wang[3], Likun Xue[4], Jianjun He[1], Wanyun Xu[1], Bing Qi[5],**
**Rongguang Du[5], Chang Liu[1], Zeyuan Li[4], Ke Gui[1], Wanting Jiang[6], Linlin Liang[1], Yan Yan[1],**
**Xiaoyan Meng[7]**
[1] State Key Laboratory of Severe Weather & Key Laboratory of Atmospheric Chemistry of CMA,
Chinese Academy of Meteorological Sciences, Beijing 100081, China
[2] Zhejiang Institute of Meteorological Science, Hangzhou 310008, China
[3] State Environmental Protection Key Laboratory of Formation and Prevention of Urban Air Pollution
Complex, Shanghai Academy of Environmental Sciences, Shanghai 200233, China
[4] Environment Research Institute, Shandong University, Ji'nan, Shandong 250100, China
[5] Hangzhou Meteorological Bureau, Hangzhou 310051, China
[6] Plateau Atmospheric and Environment Laboratory of Sichuan Province, College of Atmospheric
Science, Chengdu University of Information Technology, Chengdu 610225, China
[7] State Environmental Protection Key Laboratory of Quality Control in Environmental Monitoring,
China National Environmental Monitoring Centre, Beijing 100012, China
*Correspondence to*: Gen Zhang (zhanggen@cma.gov.cn) and Honghui Xu (forsnow@126.com)
**Abstract.** Complex aerosol and photochemical pollution (ozone and peroxyacetyl nitrate (PAN))
frequently occur in eastern China and mitigation strategies to effectively alleviate both kinds of pollution
are urgently needed. Although the effectiveness of powerful control measures implemented by the
Chinese State Council has been comprehensively evaluated on reducing atmospheric primary pollutants,
the effectiveness on mitigating photochemical pollution is less assessed and therein the underlying
mechanisms are still poorly understood. The stringent emission controls implemented from 24 August to
6 September, 2016 during the summit for Group of Twenty Finance Ministers and Central Bank
Governors (G20) provides us a unique opportunity to address this issue. Surface concentrations of
atmospheric $O_3$, PAN, and their precursors including volatile organic compounds (VOCs) and nitrogen
dioxides ($NO_x$), in addition to the other trace gases and particulate matter were measured at the National
Reference Climatological Station (NRCS) (30.22 °N, 120.17 °E, 41.7 m a.s.l) in urban Hangzhou. We
found significant decreases in atmospheric PAN, $NO_x$, the total VOCs, $PM_{2.5}$, and sulfur dioxide ($SO_2$)
under the unfavorable meteorological condition during G20 (DG20) relative to the adjacent period
before and after G20 (BG20 and AG20), indicating that the powerful control measures have taken into
effect on reducing the pollutants emissions in Hangzhou. Unlike with the other pollutants, daily
maximum average-8 h (DMA8) $O_3$ exhibited a slight increase and then decrease from BG20 to AG20,





which was mainly attributed to the variation in the solar irradiation intensity and regional transport
besides the contribution from the implement of stringent control measures. Results from
observation-based chemical model (OBM) indicated that acetaldehyde and methyl glyoxal (MGLY)
were the most important second-generation precursors of PAN, accounting for 37.3-51.6% and
22.8%-29.5% of the total production rates including the reactions of OVOCs, propagation of other
radicals, and the other minor sources. Moreover, we confirmed the productions of PAN and $O_3$ were both
sensitive to VOCs throughout the whole period, specifically dominated by aromatics in BG20 and DG20
but alkenes in AG20. These findings suggested that reducing emissions of aromatics, alkenes, and alkanes
would mitigate photochemical pollution including PAN and $O_3$. Source appointment results attribute the
reductions of VOCs source and ozone formation potentials (OFP) during G20 to the effective emission
controls on traffic (vehicle exhaust) and industrial processes (solvent utilization and industrial
manufacturing). However, fuel combustion and biogenic emission both weakened such effect with
sizeable contribution on the VOCs mixing ratios (18.8% and 20.9%) and OFPs (25.6% and 17.8%),
especially during the latter part of G20 (G20 II) when anthropogenic VOCs were substantially reduced.
This study highlights the effectiveness of stringent emission controls in relation to traffic and industrial
sources, but a coordinated program related with controlling fuel combustion and biogenic emissions is
also required on addressing secondary pollution.
**1 Introduction**
Complex atmospheric pollution including particulate and photochemical pollution (ozone ($O_3$) and
peroxyacetyl nitrate (PAN)) is a pervasive environmental issue in eastern China (Geng et al., 2007;
Ding et al., 2013; Mo et al., 2015; Li et al., 2016; Zhang et al., 2018). Numerous mitigation strategies
have been released by the Chinese government, such as the nationwide application of flue-gas
desulfurization (FGD) devices in power plants after 2006 (Feng et al., 2014) and "Atmospheric
Pollution Prevention and Control Action Plan" in 2013 (Zhang et al., 2016). As expected, ambient
concentrations of primary gas pollutants such as sulfur dioxide ($SO_2$) (Koukouli et al., 2016) and
nitrogen oxides ($NO_x = NO + NO_2$) (de Foy et al., 2016) showed good response to emission reductions.
However, secondary atmospheric pollutants such as ozone and secondary aerosols, which are dominant
compounds of fine particulate matter, frequently exceeded their respective Chinese Grade II standards
over urban cities in China (Wang et al., 2014). Severe haze pollution, mainly comprised of $PM_{2.5}$
(particles within 2.5 μm diameter range), still occur in China during wintertime, although it started to
decline during the 11[th] Five-Year Plan period (Huang et al., 2014; Cheng et al., 2016; Miao et al., 2018;
Miao and Liu, 2019). Surface $O_3$ also exhibits a rapid increasing trend over China since 2000
(Verstraeten et al., 2015; Wang et al., 2017), with high levels (9.5-14.0 ppbv) of PAN often encountered
during $O_3$ pollution events (Shao et al., 2009; Liu et al., 2010; Zhang et al., 2012a; Zhang et al., 2014;
Zhang et al., 2015; Xue et al., 2014c). Due to the highly nonlinear response of $O_3$ and PAN to primary


pollutant emissions, the mitigation of secondary photochemical pollution is even more challenging. In
the troposphere, $O_3$ and PAN are both formed in photochemical reactions of VOCs in the presence of
$NO_x$. However, PAN is exclusively formed by the oxidation of a small part of VOCs that can generate
peroxy acetyl radical ($CH_3C(O)O_2$, PA) including oxygenated VOCs (OVOCs) such as acetaldehyde,
acetone, methacrolein (MACR), methyl vinyl ketone (MVK), and methyl glyoxal (MGLY) (Williams et
al., 2000; LaFranchi et al., 2009), while $O_3$ formation involves almost all VOCs. Therefore, PAN is
considered to be a better indicator for photochemical smog than $O_3$ (McFadyen and Cape, 2005). In
addition, these OVOCs are mainly oxidation products (here referred to secondary precursors of PAN) of
a certain class of hydrocarbons (e.g., ethane, propene, isoprene, and some aromatics) by the oxidations
of $OH/NO_3/O_3$. The relative importance of individual precursors to the formation of PAN and $O_3$ varies
from place to place depending on the reactivity and composition of VOCs. Identification of the
dominant precursors is the key to effective control of photochemical pollution, which, however, remains
poorly characterized in China.
Recently, a series of temporary and stringent emission control measures were implemented in China
during several mega-events including the 29[th] Summer Olympic Games (August 2008), the 21[th]
Asia-Pacific Economic Cooperation (APEC) conference (November 2014), and China Victory Day
Parade (Victory Parade 2015) in Beijing (Verstraeten et al., 2015) and the surrounding areas (Xu et al.,
2010; Zhang et al., 2012b; Gao et al., 2011; Li et al., 2017). During these events, the effectiveness of a
series of emission control measures on reducing atmospheric primary pollutants, in particular to the
particulate matter, has been comprehensively evaluated, but the effectiveness on photochemical
pollution are less evaluated.
In September 2016, the Group of Twenty (G20) summit was hosted in Hangzhou, the capital city of
Zhejiang Province, which is located along the mid-Yangtze River Delta (YRD) in the eastern part of
China. Similar with other major events held in Beijing, rigorous temporal control measures were set to
reduce emissions of air pollutants in Hangzhou and the adjacent regions including Zhejiang, Shanghai,
Jiangsu, and Anhui province from 24 August to 7 September. These control measures included
restrictions on the number of vehicles, limited production or complete shut-down of industrial
enterprises, and temporary cessation of construction activities, and the target sources incorporated
vehicles, paint and solvent use, steel factories, chemical factories, power plants. In this study, to
evaluate the effectiveness of emission control measures on reducing pollutant concentrations, we
compared the variations of atmospheric $O_3$, PAN, particulate matter, VOCs, $NO_x$, and other trace gases
before, during, and after G20, also demonstrating the effect of meteorological conditions by using
WRF-Chem model. An observation-based chemical box model (OBM) was used to identify the
predominant precursors and key chemical processes in PAN and $O_3$ formation and to further assess the
effect of reducing their respective precursors before, during, and after G20. Positive matrix factorization
(PMF) was employed to appoint the corresponding sources of various VOCs and compare their



variations and their respective ozone formation potentials (OFPs) before, during, and after G20.

## 2. Experimental

### 2.1 Observations

In-situ observations of atmospheric PAN, $O_3$, and VOCs and a suite of associated chemical species and
meteorological parameters, including $NO_x$, CO, $SO_2$, fine particulate matter ($PM_{2.5}$), were conducted at
an urban site named as National Reference Climatological Station (NRCS) (30.22°N, 120.17°E, 41.7 m
a.s.l) in the center of Hangzhou as shown in Figure S1 in Supplement (SI). PAN was measured by a
modified gas chromatography (Agilent 7890B, USA) equipped with electron capture detector, which
has been described in our previous studies in details (Zhang et al., 2012a; Zhang et al., 2014; Zhang et
al., 2015). Trace gases including $O_3$, $SO_2$, $NO_x$, and CO were detected by a set of commercial trace gas
analyzers (Thermo Environmental Instruments Inc., USA i-series 49i, 43i, 42i, and 48i), respectively
(Zhang et al., 2018). Ambient VOCs were measured by using an on-line gas chromatography (Syntech
Spectras Instrument Co., Ltd., The Netherlands) coupled with dual detectors (Photo Ionization Detector
(PID) and flame ionization detector (FID) for quantifying $C_2$-$C_5$ VOCs (GC955 series 811) and PID for
detecting $C_6$-$C_{12}$ VOCs (GC955 series 611). Ambient $PM_{2.5}$ samples were collected using co-located
Thermo Scientific (formerly R&P) Model 1405D samplers.

### 2.2 Models

#### 2.2.1 WRF-Chem model

To quantify the separate effects of meteorological condition (EMC) and emission control measures
(EEC) on observed particulate concentrations, we performed simulations using Weather Research and
Forecasting model coupled to Chemistry (WRF-Chem). WRF-Chem V3.9 was used to simulate the
variation of $PM_{2.5}$ concentration from Aug. 6 00:00 UTC, 2016 to Sep. 16 00:00 UTC, 2016.
Multi-resolution Emission Inventory for China at 0.25° in 2016, developed by Tsinghua University
(http://www.meicmodel.org/), was used as input for WRF-Chem. WRF-Chem was configured to have
two nested domains, i.e. an outer domain with horizontal resolution of 25 km (140×100 grid points)
covering East China and the surrounding areas and an inner domain with 5 km-resolution (101×101 grid
points) covering Yangtze River Delta. Hangzhou is located in the center of domain. Vertically, there
were a total of 35 full eta levels extending to the model top at 50 hPa, with 16 levels below 2 km. The
National Centers for Environmental Prediction (NCEP) Final Operational Global Analysis (FNL) data
available at 1°×1° every six hours were used meteorological driving fields. Analysis nudging was used
for domain one. RADM2 chemical mechanism and MADE/SORGAM aerosols were used in this study.
In principle, the net contribution (NCC) of emission controls and meteorological conditions primarily
results in the difference between observed $PM_{2.5}$ before and during G20, which is represented by the
ratio of (Observed $PM_{2.5}$ (BG20)- Observed $PM_{2.5}$ (DG20 II))/Observed $PM_{2.5}$ (BG20). The effect of





meteorological conditions (EMC) was quantified by comparing the modeled $PM_{2.5}$ without emission
controls before and during G20 under their respective meteorological condition (Equation 1). Thereby,
the effect of emission controls (ECC) could be obtained through the difference between NCC and EMC
before and during G20 (Equation 2) below

$$EMC = \frac{Modeled\ PM_{2.5}\ (BG20) - Modeled\ PM_{2.5}\ (DG20\ II)}{Modeled\ PM_{2.5}(BG20)} \times 100\% \qquad (1)$$

$$ECC = (NCC - EMC) \times 100\% \qquad (2)$$

In general, the modeled results of $PM_{2.5}$ before and after G20 can reproduce the observation results
(mean bias (MB) =2.46, root mean-square error (RMSE) = 15.5, R = 0.63, p < 0.01), providing the basis
of the following comparison.

### 2.2.2 Backward trajectories analysis

To determine the influence of regional transport on the pollutant concentrations, 24 h air mass back
trajectories starting at 300 m from NRCS site were calculated by using the National Oceanic and
Atmospheric Administration (NOAA) HYSPLIT-4 model with a $1^{o} \times 1^{o}$ grid and the final meteorological
database. The 6-hourly final archive data were obtained from the National Center for Environmental
Prediction's Global Data Assimilation System (GDAS) wind field reanalysis. GDAS uses a spectral
medium-range    forecast    model.    More    details    can    be    found    at    http://www.arl.
noaa.gov/ready/open/hysplit4.html. The model was run 24 times per day. The method used in trajectory
clustering was based on the GIS-based software TrajStat (Wang et al., 2004).

### 2.2.3 Observation-based chemical box model (OBM)

Here we used OBM model to simulate in situ PAN and $O_3$ production and their sensitivity to changes in
PAN and $O_3$ precursors, which has been successfully implied in our previous studies (Xue et al., 2014a;
Xue et al., 2014c; Xue et al., 2016; Li et al., 2018). In brief, the model was built on the latest version of
the Master Chemical Mechanism (MCM v3.3), an explicit mechanism describing the degradation of 143
primarily emitted VOC, resulting in 17,224 reactions involving 5833 molecular and free radical species
(Saunders et al., 2003). Besides the existing reactions in MCM v3.3, the heterogeneous reactions of
$NO_2$, $HO_2$, $NO_3$, and $N_2O_5$ were also incorporated. In addition, we also optimized the model with some
physical processes such as the variations of boundary layer height and solar zenith angle, dry deposition,
and the dilution of air pollutants within the planetary boundary layer (Xue et al., 2014b). The photolysis
frequencies appropriate for Hangzhou are parameterized using a two-stream isotropic-scattering model
under clear sky conditions. In this study, all of these reactions were tracked and grouped into a small
number of formation pathways, such as acetaldehyde, acetone, MACR, MVK, MGLY, other OVOCs,
reactions of $O_3$ with isoprene and MPAN, and propagation of other radicals to PA. The production rate
of PA could be estimated as the sum of these reaction rates. The ozone production rates were calculated



through the oxidation of NO by $HO_2$ and $RO_2$, and its destruction rates were mainly facilitated by $O_3$
photolysis and reaction with NO, $NO_2$, OH, $HO_2$, and unsaturated VOCs. Moreover, we investigated the
sensitivities of PAN and $O_3$ formation to their respective precursor species by introducing a relative
incremental reactivity (RIR) concept which is widely applied in the OBM investigation of PAN and
ozone formation (Chameides et al., 1999; Xue et al., 2014c). In this calculation, we performed model
calculations during the period of 20 August-10 September, 2016, during which the VOCs measurement
were available. The model was run based on the hourly average profiles of PAN, $O_3$, CO, $SO_2$, NO, $NO_2$,
$C_2$-$C_{10}$ NMHCs, air temperature and pressure, and RH measured at NRCS site. During the simulation,
the model was pre-run for three days with constrain of the data of 20-22 August so that it reached a
steady state for the unmeasured species (e.g., MACR, MVK, HONO, radicals). More detailed
description of this model has been given in previous studies (Jenkin et al., 2003; Xue et al., 2014a; Xue
et al., 2014c).

### 2.2.4 Positive matrix factorization (PMF) Model


Positive matrix factorization (PMF) is an effective source apportionment receptor model based on the
fingerprints of the sources that does not require the source profiles prior to analysis and has no
limitation on source numbers (Hopke, 2003; Pentti and Unto, 1994). The data used in PMF is of the
form of an i×j matrix X, in which i is the sampling number and j is the number of species. Based on
chemical mass balance of the pollutants, the following equation can be derived as:

$$X_{ij} = \sum_{k=1}^{p} g_{ik}f_{ik} + e_{ij}$$

where p is the number of the sources (i.e., the number of factors), f is the profile of each source, g refers
to the contribution of each factor to the total concentration, and e is the residual. Factor contributions
and profiles are derived by minimizing the total scaled residual Q:

$$Q = \sum_{i=1}^{n} \sum_{j=1}^{m} \left(\frac{e_{ij}}{u_{ij}}\right)^2$$

where u is the uncertainty of the sampling data. More details about principles have been found
elsewhere (Cai et al., 2010; Zhang et al., 2013; Li et al., 2017; Li et al., 2015). In this study, we used
EPA PMF 5.0 model to identify major VOCs sources and their temporal variations. We discarded the
species that were below MDL for more than 50% of the time or showed a significantly smaller signal to
noise ratio (S/N). The uncertainties for each sample and species were calculated based on the following
equation if the concentration is greater than the method detection limit (MDL) provided:

$$\text{Uncertainty} = \sqrt{(0.5 \times \text{DML})^2 + (\text{Error Fraction} \times \text{Concentration})^2}$$

Values below the detection limit were replaced by one-half of the MDL and their overall uncertainties
were set at five-sixths of the MDL values. In this analysis, different numbers of factors were tested. The
robust mode was used to reduce the influence of extreme values on the PMF solution. More than 95% of





the residuals were between -3 and 3 for all compounds. The Q values in the robust mode were
approximately equal to the degrees of freedom.

## 3 Results and discussion

In order to comprehensively evaluate air quality during the G20 period, we compared the concentrations
of pollutants during G20 with the adjacent time period in 2016, respectively. According to the control
measures schemes, we classified the whole period into three episodes: one week before G20 (BG20)
(16-23 August, 2016), during G20 (DG20) (24 August-6 September) including Phase I (24-27 August)
and Phase II (28 August-6 September), and one week after G20 (AG20) (7-15 September). During
phase I the government implemented strict emission control measures in industrial source, power plant,
and residential and the phase II referred to the additional controls measures as vehicles controls in the
Hangzhou and surrounding provinces (including Zhejiang, Jiangsu, Jiangxi, and Anhui).

### 3.1 Evolutions of meteorological condition

First, we looked into the day-to-day variations of meteorological parameters and atmospheric pollutants
from BG20 to AG20 in Fig. S2 in SI. In the period of BG20 and the beginning of DG20 I (16-25
August), subtropical anticyclone dominated the Hangzhou and surrounding area, leading to continuous
10 days with daily mean temperature of 31.5 $^{o}$C ranged from 29.9-32.5 $^{o}$C and strong solar irradiation
intensity (mean daily maximum value: 369.4 W m$^{-2}$), favorable for the photochemical production of O$_3$
and PAN. The highest O$_3$ (113.4 ppbv) occurred at 13:00 LT on 25 August under the maximum air
temperature of 35.2 $^{o}$C. Meanwhile, the mean daily maximum height of mixing boundary layer (MBL)
during this period was up to ca. 1895 m, beneficial for the diffusion of atmospheric primary pollutants
in the vertical direction. In addition, the prevailing wind was from east (15.1%) with a mean wind speed
of 2.9 m s$^{-1}$. Results from the backward trajectory simulations demonstrated that the air masses from the
east originated from the East China Sea and Yellow Sea, bringing in clean marine air (Fig. S3). Thus,
meteorological conditions before G20 were favorable for the dispersal of atmospheric pollutants. On 26
and 27 August, the weather pattern changed to a cold continental high with showery and windy days.
The total precipitation and mean wind speed both reached their respective maximums of 14.6 mm and
3.7 m s$^{-1}$ on 26 August. Accordingly, all species except CO significantly decreased by 12.3% for SO$_2$,
29.7% for NO$_x$, 6.7% for PM$_{2.5}$, 11.9% for daily maximum average-8 h (DMA8) O$_3$, and 56.1% for
PAN relative to BG20. With respect to the last half of DG20 I and the beginning of DG20 II (28-31
August), the prevailing wind experienced a shift from northwest to west and to southwest. On 28
August, the prevailed wind was from the north with the average daily maximum wind speed of 3.9 m s$^{-1}$
during G20, and the relative humidity rapidly decreased by 26.2% relative to the previous day. As seen
in Fig. S3, air masses arrived at Hangzhou from the north passed through all of Jiangsu Province and
northern parts of Zhejiang Province, two of the most developed provinces in China, with intense human



activities. They carried higher $PM_{2.5}$, $SO_2$, $NO_x$, and CO loadings than the other clusters (See Table S1).
On 1 September, the prevailing wind was from southwest with high wind speeds (3.3 m s$^{-1}$). Results
from back trajectories indicated that the southwesterly air masses originated from northern Jiangxi
Province, transported over western Zhejiang Province, and arrived at Hangzhou, with high
concentration loadings of $SO_2$, particulate matter, $O_3$, and PAN. The increased relative humidity (56.5%)
relative to 49.5% on 31 August was beneficial for the formation of particulate matter. During 2-4
September, Hangzhou area witnessed a stable meteorological condition with weak wind (WS < 2.6 m/s),
continuously high air temperature (daily maximum average: 32.2 $^o$C), and moderate relative humidity
(ca. 60%). Such condition was favorable for the accumulation of particulate matter and the
photochemical production of $O_3$. It caused significant increases by 25.1% for $PM_{2.5}$, 16.7% for $PM_{10}$,
and 10.7% for $O_3$ compared with BG20, in contrast to the large decreases by 56.4% for $SO_2$ and 27.9%
for $NO_x$ due to the implement of emission control measures. Overall, the meteorological condition
during G20 II was not favorable for the dispersal of atmospheric primary pollutants but beneficial for
producing $O_3$. However, with the proceeding of the stringent control measures, the most distinct drops
of pollutants concentrations were found on 5 September, with the large reductions of 50.0% for $PM_{2.5}$,
18.3% for DMA8 $O_3$, 55.7% for $SO_2$, 41.3% for $NO_x$, and 65.6% for PAN relative to BG20, respectively.
Within AG20, 7 rainy days with mean daily total precipitation of 18.7 mm occurred as well as 6 days
with low wind speed (ca. 2.0 m/s) and 8 days with low MBL (<1000 m). Such meteorological condition
was beneficial for scavenging the particulate matter and $SO_2$ by wet deposition in addition to the
accumulation of $NO_x$. In addition, weak solar irradiation intensity was not favorable for the
photochemical formation of $O_3$ and PAN. On 7 September a moderate showery lasted from 2:00 LT to
11:00 LT with daily total precipitation of 9.5 mm, accompanied by low air temperature (21.5 $^o$C) and
wind speed (1.8 m/s). Compared with the previous day, significant decreases of DMA8 $O_3$ (22.6%) was
found as expected, while together with a small reduction ratio of $PM_{2.5}$ (2.7%) and unexpected increases
for $NO_x$ (41.1%) and $SO_2$ (175.1%), indicating that emissions immediately bounced back after lifting
the ban on emission controls.
**3.2 Evolutions of pollutant concentrations**
Statistically, observed daytime concentrations of $PM_{2.5}$, $NO_x$, and $SO_2$ in DG20 II both exhibited
significant decreases relative to those in BG20 with the reduction ratios of 11.3%, 17.0%, and 18.0%,
respectively (Fig. 1). Furthermore, by using WRF-Chem model we quantified the contributions of the
emission control measures (ECC) with 63.5%, 44.1%, and 31.2% to the reductions of $PM_{2.5}$, $SO_2$, and
$NO_2$ in DG20 II relative to BG20, respectively, but for the meteorological conditions it made negative
contributions. This evidence well indicated that powerful control measures have taken into effect on
reducing pollutant emissions in Hangzhou under the unfavorable meteorological conditions. The large
decreases of $NO_x$ and $SO_2$ reflected the reduction of vehicle exhaust and coal consumption during G20





in Hangzhou and surrounding areas. It is worth noting that CO showed gradual increases (ca. 20.7%)
from BG20 to DG20. Fuel combustions, mainly including residential usage and liquid natural gas and
petroleum gas, around YRD regions during this period might account for such unique pattern of CO.
Under the same stringent control measures, the variation of $O_3$ was not consistent with the primary
pollutants. Observed DMA8 $O_3$ increased by 12.4% in DG20 I relative to BG20, which was attributed
to regional transport from the northern provinces and the enhanced solar radiation intensity. Afterwards,
DMA8 $O_3$ decreased by 33.4% from DG20 II to AG20 (Fig. 1), as did the peak values of mean daily $O_3$
in DG20 II compared to BG20 and DG20 I (Fig. S4). This evidence suggests that additional vehicles
controls implemented during DG20 II might have played an important role in reducing atmospheric $O_3$
pollution in Hangzhou reflected by shaping such unique diurnal variation, which was also confirmed by
the decreased OFP from vehicle exhaust below. Elevated $O_3$ during DG20 rush hours (as shown in Fig.
S2 and S3) was attributed to the reduced titration of fresh NO emission under the control measures on
vehicle exhaust. Considering such effects, $O_x$ (represented by the sum of $O_3+NO_2$) was used to
determine the local photochemical formation. The variation of DMA8 $O_x$ was similar with $O_3$, with
distinct decreasing trend from DG20 II to AG20. For PAN, it showed different pattern with $O_3$. Daytime
PAN exhibited significant decrease (ca. 45.4%) found from BG20 to DG20 II and then it sharply built
up to similar magnitudes in AG20 with BG20. Thereby, it both indicates the significant effectiveness of
emission control measures on reducing local photochemical formation of $O_3$ and PAN. The underlying
formation mechanisms of PAN and $O_3$ including their respective key precursors and chemical process
are elucidated in Sect.3.3.
With respect to VOCs, the mixing ratios of total VOCs also showed significant reduction of 20.0% in
DG20 compared with BG20, but increased by 104.1% in AG20 after control (Table S2). Alkanes were
the most abundant VOCs group (55.4%) in all periods, and were reduced by 19.8% from BG20 to DG20.
On the contrary, alkenes increased by 20.0% in DG20 compared to BG20, among which ethylene
accounted for 63.9%-78.0% during the three periods, although other alkenes decreased to a minor extent.
As expected, aromatics were reduced by 49.7% in DG20 compared with BG20. Ambient mixing ratios
of specific VOCs at NRCS station are summarized in Table S3. Ethane, ethylene, benzene, and toluene
are the four most abundant species during all the periods. Compared with BG20, except ethane,
isopentane, and ethylene, the mixing ratios of all species decreased in DG20. Ethylene, as a
representative tracer of fuel combustion, showed continuous increase from BG20 to AG20, possibly
indicating the ineffectiveness of control measures in this source.

**3.3 Identification of the Key Precursors and Chemical Processes for PAN and $O_3$**

To identify the key precursors and chemical processes for PAN, we employed the observation-based
model to investigate the daytime average contributions to PA radical production rates directly from
individual pathways for these four episodes (Fig. 2). Acetaldehyde (e.g., oxidation of OH and $NO_3$) and





MGLY (e.g., photolysis and oxidation by OH and $NO_3$) were the most important sources of PA in
Hangzhou, accounting for 37.3-51.6% and 22.8%-29.5% of the total production rates. This was in
agreement with the findings obtained from the other typical urban areas such as Beijing (Xue et al.,
2014c; Liu et al., 2010; Zhang et al., 2015), Tokyo (Kondo et al., 2008), Houston, Nashville (Roberts et
al., 2001), and Sacramento (LaFranchi et al., 2009). Reactions of OVOCs and propagation of other
radicals to PA (mainly including decomposition of some RO radicals and reactions of some higher acyl
peroxy radicals with NO) were also significant sources, with average contributions of 7.1%-9.1% and
18.1%-27.0%, respectively. A minor contribution (~1% in total) was originated from the other pathways
of $O_3$+isoprene, $O_3$+MPAN, acetone, and MVK. Acetaldehyde and other OVOCs are mainly
photooxidation products of hydrocarbons, thus it's necessary to further identify the first-generation
precursors of PAN here. We tested the model sensitivity by introducing the concept of relative
incremental reactivity (RIR), which is widely used in the OBM study of ozone formation (Chameides et
al., 1999). Here RIR is defined as the ratio of decrease in PAN production rates to decrease in precursor
concentrations (e.g., 20% reduction is used in this study). A number of sensitivity model runs were
performed to calculate the RIRs for $NO_x$, alkanes, alkenes, and aromatics classes as well as the
individual $C_2$-$C_{10}$ hydrocarbon species. As shown in Fig. 3a, production of PAN was sensitive to VOCs
from BG20 to AG20. Meanwhile, the negative RIR values for $NO_x$ also indicated a VOCs regime of
PAN production around the G20 period in urban Hangzhou. In terms of BVOCs, the positive RIRs
values for isoprene (0.18-0.38) from BG20 to AG20 implied that in-situ formation of PAN at NRCS was
highly sensitive to isoprene. As to AVOCs, alkenes and aromatics were the most important
first-generation PAN precursors, with the RIRs range of 0.24-0.37 and 0.26-0.52, respectively.
Furthermore, we identified the other specific VOCs controlling PAN production, which were xylenes,
trans/cis-2-butenes, trimethylbenzenes, toluene, and propene evidenced by their positive RIRs.
Compared with their individual RIRs between control and non-control period, the in-situ production of
PAN was dominated by aromatics in BG20 and DG20 I, but controlled by alkenes in AG20. Besides
secondary acetaldehyde formed by the oxidation of ethanol, most aromatics were mainly emitted by
vehicle exhaust. The decreased RIRs of aromatics together with the decreased contribution ratios of
acetaldehyde to the PA radical formation during G20 both indicated the effectiveness of control
measures on vehicle exhaust on reducing atmospheric PAN concentration. Similar with PAN, the
daytime average RIRs for major groups of $O_3$ precursors during the episodes are shown in Fig. 3b.
Overall, the in-situ ozone formation was also controlled by VOCs from BG20 to AG20. AVOCs were
dominated by alkenes and aromatics, along with their increasing and decreasing RIRs, respectively.
With the proceeding of emission control, the RIR for AVOCs showed gradual decrease from BG20 to
DG20, but increased after G20. In contrast, BVOCs (mainly as isoprene) exhibited gradual increases for
all periods, especially during the phase II in DG20 and AG20 when their RIRs were both higher than
those for AVOCs. Thereby, the contribution of BVOCs to the photochemical production of $O_3$ weakened





the effect of stringent control measures on reducing surface $O_3$. The RIRs for $NO_x$ were negative
throughout the period of G20, also indicating a VOC-limited regime for the sensitivity of ozone
formation. This suggests that reducing emissions of aromatics, alkenes, and alkanes would alleviate the
$O_3$ formation, yet cutting $NO_x$ emissions may aggravate the local $O_3$ problems.

**3.4 Identification of VOCs sources and quantification of their respective ozone formation potential**

To distinguish the various sources of VOCs, we compared the PMF profiles with the reference profiles
from the literature as listed below. Seven sources were identified as follows: (1) gasoline evaporation (2)
solvent utilization (3) industrial manufacturing (4) industrial chemical feedstock (5) vehicle exhaust (6)
fuel combustion (7) biogenic emission. Figure 4 exhibited the modelled source profiles together with
the relative contributions of each sources to individual species. The first source is characterized by a
significant amount (78.5%) of isopentane which is a typical tracer for gasoline evaporation (Liu et al.,
2008). Therefore, this source was identified as gasoline evaporation. The second source was rich in
n-pentane and aromatics. Many aromatics such as BTEX are the dominant components of organic paints,
and were regarded as chemical tracers of solvent utilization (Watson et al., 2001). Significant amounts
of ethylbenzene, xylenes, and n-pentane present in the second source, accounting for 19.2%, 58.8%, and
98.8%, respectively. Thus, the second source was identified as solvent utilization. The third source was
characterized by high loading of cyclohexane (54.7%) and BTEX (15.1%-46.2%). These compounds
are confirmed to be typical species in the industrial manufacturing in China (Liu et al., 2008). Thus, this
source was representative of industrial manufacturing. The fourth source identified as industrial
chemical feedstock (shown in Fig. 4) was characterized by a very little contribution to alkanes and
aromatics and   large amounts of 3-ethyltoluene (29.4%), 3-methylheptane (51.0%), and n-hexane
(47.1%), which are typical proxies for industrial chemical feedstock (Liu et al., 2008; Mo et al., 2015).
The fifth source was characterized by abundant 2-methylpentane (61.7%) and BTEX, which is a typical
tracer for vehicle exhaust (Liu et al., 2008; Li et al., 2015). In addition, 2, 2, 4-trimethylpentane is a fuel
additive used to gain higher octane ratings (McCarthy et al., 2013) with high abundance of 21.4% in
this source and thus it is identified as vehicle exhaust. The sixth source profile shown in Fig. 4 was in
relation to 48.9% of the total measured ethylene mixing ratios, of which was major species emitted from
fuel combustion process (Li et al., 2015). It was also characterized by significant amounts of ethane,
propane, n-butane, propene, and benzene. Ethane and propane are the tracers of natural gas and liquid
petroleum gas (LPG) usage, respectively, and the source profiles of resident fuel combustion in China
contained alkenes (Wang et al., 2013). Coal combustion can release a large amount of BTEX into the
atmosphere and styrene is a typical indictor of industrial manufacturing in China (Liu et al., 2008; Li et
al., 2015). Thus, this source was believed to be as fuel combustion related with industrial process and
residual usage. The seventh source was distinguished by a significant amount of isoprene, a





representative indicator of biogenic emission. About 93.1% of the total isoprene mixing ratios is apportioned to this factor (Guenther et al., 1995). There were very small quantities of the other species such as aromatics (0-1.8%) in this factor. Therefore, it was excluded from biomass burning but mainly identified as biogenic emission. Figure 5 shows the variation of the seven sources during the four periods. Clearly, anthropogenic sources such as solvent utilization, industrial manufacturing, vehicle exhaust, fuel combustion, and industrial chemical feedstock were the predominant sources to the total VOCs before and after G20, as high as 52.4%-81.7%. Furthermore, anthropogenic emission showed significant reductions during G20 response to the stringent emission control. In BG20, solvent utilization was the predominant contributors to VOCs mixing ratios, contributing 1.88 ppbv, followed by vehicle exhaust (1.77 ppbv, 21.6%), industrial manufacturing (1.55 ppbv, 19.0%), biogenic emission (1.16 ppbv, 14.1%), gasoline evaporation (0.83 ppbv, 10.1%), and fuel combustion (0.35 ppbv, 4.3%). The industry-related emission (industrial manufacturing, chemical feedstock, and solvent utilization) together accounted for 50.0% of the total VOCs mixing ratios. The vehicle-related emission sources (vehicle exhaust and gasoline evaporation), accounted for 31.7% of the total VOCs mixing ratios. It indicated that traffic and industry sources were the major VOCs sources before the control period. Compared with BG20, the contribution of solvent utilization was reduced to the largest extent, with a magnitude of 1.43 ppbv, followed by industrial manufacturing (0.69 ppbv), and vehicle exhaust (0.38 ppbv), during the first emission control period (DG20 I). According to the control strategy during G20, the control measures of source emission were mainly on the industry and power plant in DG20 I, and thus it was responsible for the large reduction of industry-related emission including solvent utilization (76.0%), industrial manufacturing (44.0%), and vehicle exhaust (21.0%). With the acceleration of emission control (DG20 II), the contribution of vehicle-related emission was reduced as expected in vehicle exhaust (66.1%) and gasoline evaporation (61.8%) relative to DG20 I, while significant increase was also found in fuel combustion with the increment of 0.7 ppbv (152.6%). After G20, the contributions of vehicle-related emission and industry-related emission both showed bounces due to lifting a ban on industry, power plant, and transport in and around Zhejiang Province. It should be mentioned that biogenic emission also played an indispensable importance in contributing to the VOCs mixing ratios, from 0.81 ppbv to 1.29 ppbv. About 20.9% of the total VOCs mixing ratios could be ascribed to the biogenic emission, acting as the second major source, during the G20 II period. It indicated that biogenic VOCs might make more contribution to the VOCs mixing ratios especially when anthropogenic VOCs were substantially reduced following the process of control measures.

Moreover, we quantified their respective ozone formation potential (OFP) before, during, and after G20 by using the latest maximum incremental reactivity (MIR) and the appointed concentration profiles above (See Fig. 6). Overall, the total OFP in DG20 was significantly reduced by the implement of stringent control measures compared with BG20 and AG20. Specifically, the OFPs of solvent utilization, industrial manufacturing, and vehicle exhaust both showed significant decreases (17.3%-77.2%)





compared with BG20, while fuel combustion significantly increased by 52.2% with the OFP of 6.9 ppbv,
accounting for 25.6% of the total during G20. Thus, it is clear that the high OFP of fuel combustion
contributed by ethylene was also responsible for the enhanced concentration of $O_3$ during G20. Such
high OFP from fuel combustion was also elucidated in APEC in Beijing (Li et al., 2015). To classify the
specific fuel type, we first examined the fire spots derived from the Fire Inventory NCAR Version-1.5
(FINNV1.5) in eastern China before, during, and after the period of 2016 G20 (See Fig. S5 in SI). Straw
combustion was excluded according to the decrease in the number of fire spots in the same time period
from BG20 to AG20. As mentioned above, industrial process with coal combustion was strictly limited
throughout the whole G20 period. To ensure the clean energy used in 2016 G20, local government
accelerated the supply of liquid natural gas during the 13[th] Five-Year Plan period in Hangzhou. In 2016,
the consumption amounts of natural gas and liquid petroleum gas both increased up to $4.55 \times 10^9$ kg
(54.4%) and $5.09 \times 10^8$ kg (13.4%) compared with those in 2015, respectively (ZPSY, 2016, 2017). Thus,
liquid natural gas and petroleum gas were identified as the major fuel used in the residential usage
during G20. After G20, all anthropogenic sources both showed significant increments of OFP, among
which the fastest growth of source was vehicle exhaust (17.6 ppbv, 638.4%), followed by fuel
combustion (9.4 ppbv, 35.1%), industrial manufacturing (7.7 ppbv, 89.2%), and solvent utilization (7.4
ppbv, 258.1%), respectively.

## 4 Conclusions

In this study, ground-based concentrations of atmospheric trace gases and particulate matter, together
with meteorological parameters, were measured at a NRCS site in urban Hangzhou before, during, and
after G20. We found significant decreases in atmospheric VOCs, $PM_{2.5}$, $NO_x$, and $SO_2$ in DG20 relative
to BG20 and AG20, respectively, under the unfavorable meteorological conditions (e.g., stable weather
pattern and regional transport). This evidence well indicated that the powerful control measures have
taken effect in their emissions in Hangzhou. On the contrary, observed DMA8 $O_3$ increased from BG20
to DG20 I, which was attributed to the regional transport from the northern provinces and the enhanced
solar radiation intensity, and then decreased from DG20 II to AG20. The decreases in the peak
concentration of daily $O_3$ and the OFP estimated from various VOCs sources both suggested the
effectiveness of stringent control measures on reducing atmospheric $O_3$ concentrations. Unlike $O_3$, PAN
exhibited gradual decrease from BG20 to DG20. With the OBM model, we found acetaldehyde and
methyl glyoxal (MGLY) to be the most important second-generation precursors of PAN, accounting for
37.3-51.6% and 22.8%-29.5% of the total production rates including the reactions of OVOCs,
propagation of other radicals, and other minor sources. Furthermore, we confirmed that the production
of PAN was sensitive to anthropogenic and biogenic VOCs (isoprene) throughout the whole period,
specifically aromatics in BG20 and DG20 I but alkenes in AG20. Similarly, the sensitivity of ozone
formation was also under VOC-limited regime throughout G20 period. These findings suggest that




reducing emissions of alkanes, alkenes, and aromatics would mitigate photochemical smog including
PAN and $O_3$ formation. Furthermore, traffic (vehicle exhaust and gasoline evaporation) and industrial
sources (solvent utilization, industrial manufacturing, and chemical feedstock) were found to be the
major VOCs sources before G20, accounting for ca. 50.0% and 31.7% of the total, respectively, with the
ozone formation potential (OFP) of 14.4 ppbv and 16.1 ppbv. Large decreases were found in the sources
and OFPs of solvent utilization (74.1% and 17.3%), followed by vehicle exhaust (57.4% and 77.2%)
and industrial manufacturing (56.0% and 40.3%) response to the stringent control measures during G20,
but significantly increased by 4.2 and 2.6, 0.7 and 6.4, and 1.7 and 0.9 times after G20 due to lifting a
ban on industry, power plant, and transport in and around Zhejiang Province. We also appeal to pay
attention on controlling fuel combustion and biogenic emission especially when anthropogenic VOCs
were substantially reduced following the process of control measures. The experience of G20 suggests
that stringent emission controls do effectively address primary pollution, but a coordinated program
related with controlling fuel combustion and biogenic emissions is required to mitigate secondary
pollution.
***Author contributions.*** GZ and HX designed research; HW, BQ, RD, and XM performed research, GZ,
LX, JH, WX, CL, LL, ZL, KG, YY, and WJ analyzed data; and GZ, HX, LX wrote the paper.
Data

***Data availability.*** The data in the figures in both the main text and the Supplement are available upon
request to the corresponding author (Gen Zhang, zhanggen@cma.gov.cn).

***Competing interests.*** The authors declare that they have no conflict of interest.

***Acknowledgements.*** This study is financially supported by National Key Research and Development
Program of China (2016YFC0202300), National Natural Science Foundation of China (41775127 and
41505108), State Environmental Protection Key Laboratory of the Cause and Prevention of Urban Air
Pollution Complex (Y201701), and Zhejiang Provincial National Science Foundation (LY19D050002).
The authors are especially grateful to Dr. Xiaobin Xu for the help in discussions.



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





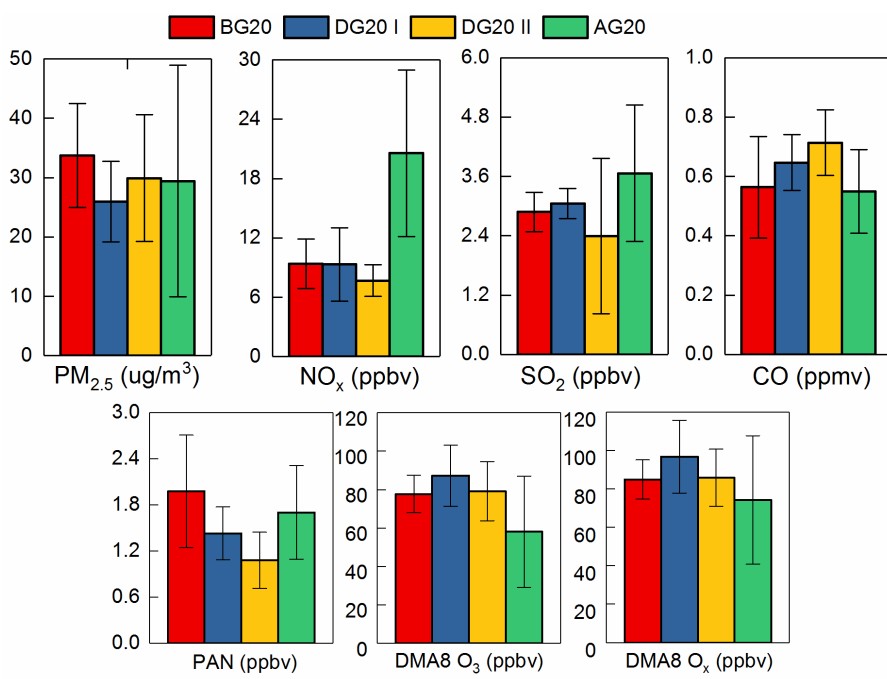


Figure 1. The comparisons of daytime PM$_{2.5}$, NO$_x$, SO$_2$, CO, PAN, DMA8 O$_3$, and DMA8 O$_x$. before,

during, and after G20, denoted as BG20, DG20, and AG20, respectively. The error bars represent the

standard deviations.




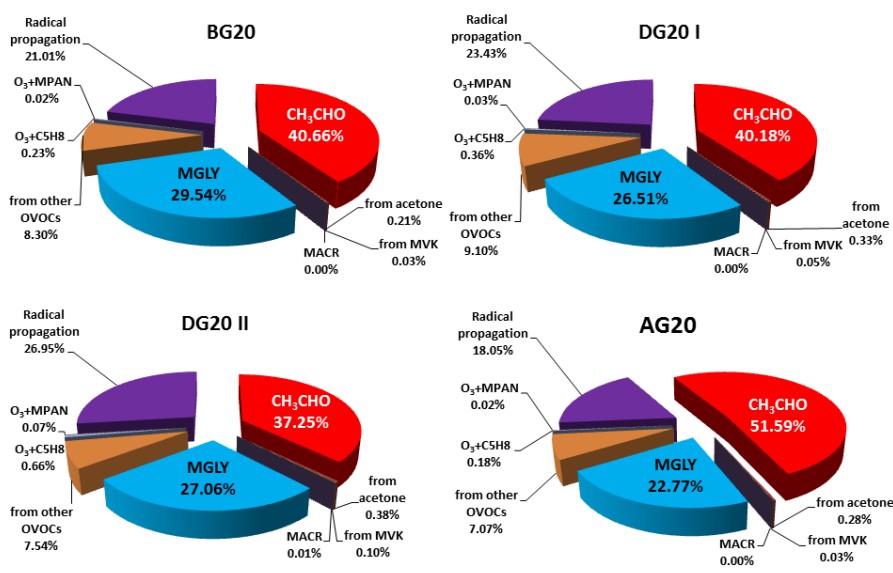


Figure 2. Contributions of individual pathways to PA radical formation during the episodes of BG20, DG20 I, DG20 II, and AG20, respectively.




Figure 3. Sensitivity of PAN (a) and $O_3$ (b) production rate to major precursor groups and individual species (09:00-17:00). Error bars are standard deviations.

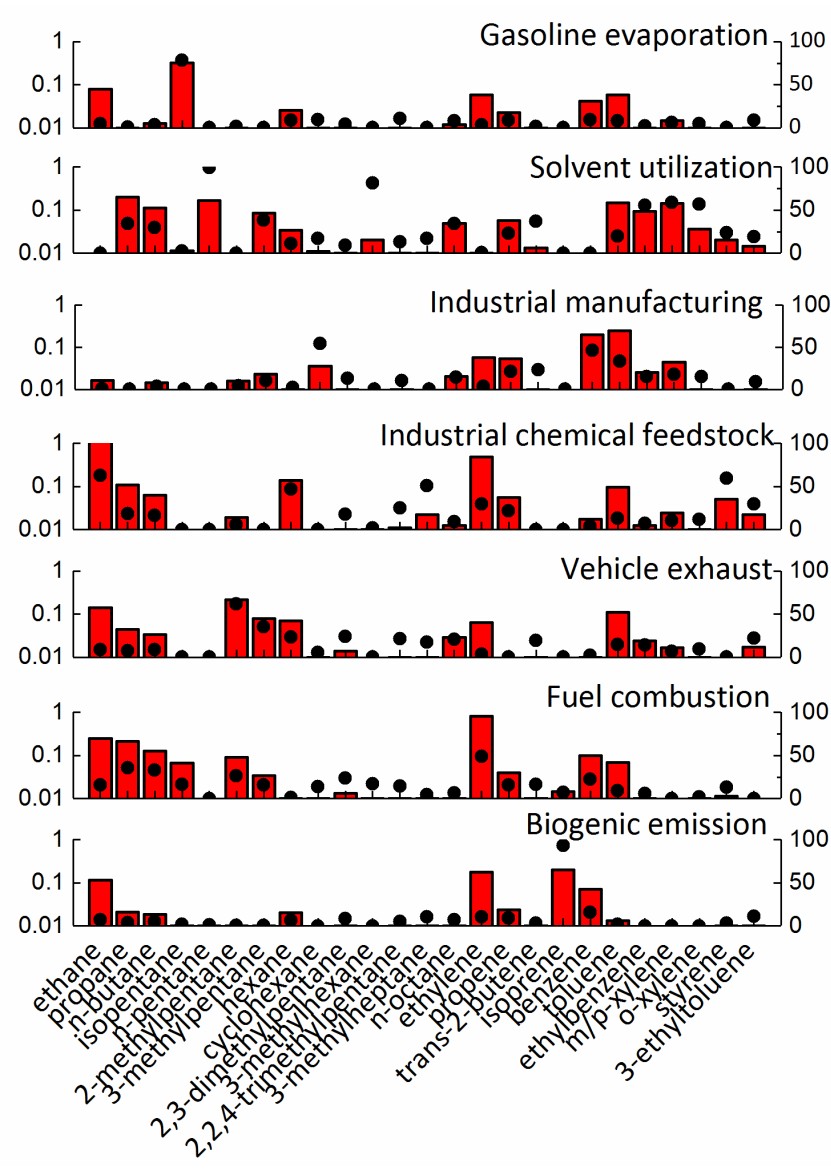


Figure 4. Seven source profiles and their respective contribution resolved from PMF model. The bars
are the profiles (ppbv, left axis), and the dots are the percentage contribution (%, right axis) from
individual factor.




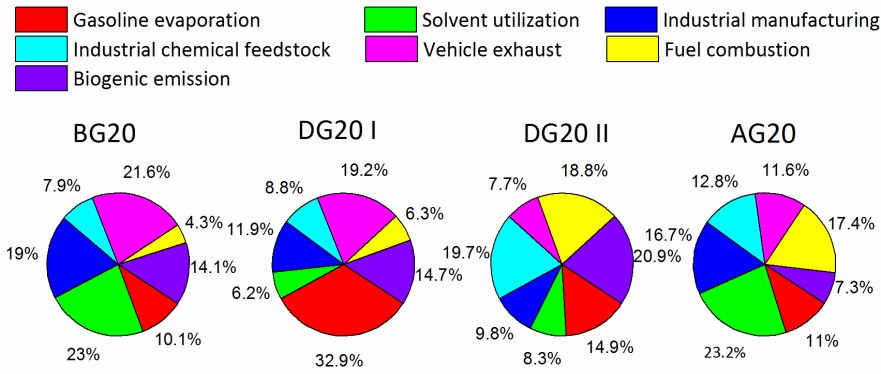


641            Figure 5. Variation of the sources (percentage) during the four periods






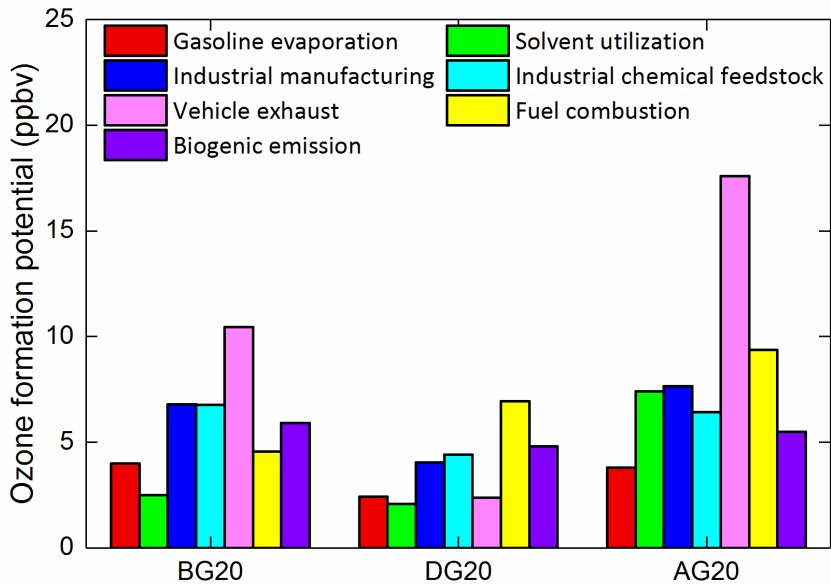


Figure 6. Ozone formation potential (ppbv) of each source before, during, and after the control period during 2016 G20 in China