# Peer review of "Exploring the inconsistent variations in atmospheric primary and secondary pollutants during the G20 2016 Summit in Hangzhou, China: 2 implications from observation and model 3 Gen Zhang1\*, Honghui Xu2\*, Hongli Wang3, Likun Xue4, Jianjun He1,"

_Atmospheric Chemistry and Physics, 2019_

## Referee Comment (RC1) · Anonymous Referee #2 · 14 Jan 2020

Based on the unique case of G20 held in Hangzhou, the authors systematically evaluated the effectiveness of powerful control measures implemented by the Chinese State Council on reducing atmospheric primary (i.e., NOx, SO2, and CO)and secondary pollutants(PAN and O3) after discriminating the effect of meteorological condition during G20. Then, they explored the underlying mechanisms of photochemical pollution including PAN and O3 by using MCM, appointed the source of VOCs by PMF model and further calculated the OFP for these various sources. The observational dataset are valuable, and the manuscript reports the measurement results well. In summary,

the topic is very interesting andthe manuscriptisalso of good quality. Thus I strongly recommend it could be published in the ACP after minor revision below. 1) line25-26 reconstruct this sentence as ". . .during G20 Summit provide us a unique opportunity toaddress this issue. Surfaceconcentrations of. . ." 2) line 53 add the phase "matter" after "particulate" 3) line 89-90 rewrite this sentence 4) line 120-121 add some detailed information about the PM2.5 measurement. 5) line 124-125 the abbreviate phase of "EMC" and "EEC" are not consistent with those below. Revise them.

---

## Referee Comment (RC2) · Anonymous Referee #3 · 27 Jan 2020

The authors evaluated the effectiveness of pollution control measures implemented during the G20 2016 Summit in Hangzhou, China. Field observation on NOx, SO2, CO, VOCs, PM10, PM2.5, PANïijŇand O3 were carried out. OBM and PMF model tools were used to analyze the data. It's valuable to publish in this journal. However, the English writing should be improved before publication.

Specific comments:

Line 269-270: CO showed a gradual increase ($\sim$20.7%), which is not consistent with

[Figure]

SO2, NO2, and PM. It seems that CO sources are very different with NOx and SO2 sources in Hangzhou or pollution controls are not effective on CO reduction. Could the authors give more explanations? I also notice that 48i analyzer is used for the measurement. As we know, zero drift is inevitable for this kind of principle. So, pls provide the quality control measures during the observation.

In Fig. 1, TVOCs is needed to add.

It seems that PM10 and PM2.5 results play no roles on the data analysis in the whole context.

Fig. S1 is better in the manuscript than in the supplement information.

Fig.5, Similar fuel combustion contributions are found in DG20-II and AG20, which is very different with that in BG20. Why?

Much more contents are done in section 3.4 (VOCs source identification and OFP quantification). How do those results relate with the inconsistent variations in the primary and secondary pollutants?
* * *

---

## Referee Comment (RC3) · Anonymous Referee #1 · 29 Jan 2020

The manuscript describes a comprehensive observational dataset including atmospheric O3, PAN, particulate matter, VOCs, NOx, and other trace gases to evaluate the effectiveness of emission control measures on reducing pollutant concentrations before, during, and after G20. It's very reasonable to demonstrate the effect of meteorological conditions by using WRF-Chem model. Further, an explicit OBM model was used to identify the predominant VOCs precursors and key chemical processes in PAN and O3 formation and to further appoint the corresponding VOCs sources before, during, and after G20 by using PMF model. The manuscript is clearly written and for-

matted very well. Thus, after considering several comments below as minor revisions, I recommend the publication of this manuscript in ACP. 1. The authors mentioned emission control measures contributed 63.5%, 44.1% and 31.2% to the reductions of PM2.5, SO2 and NO2 in DG20 II relative to BG 20. And meteorological conditions made negative contributions. What are the other factors contributing to the reduction of the observed pollutants. 2. What are the contribution of emission control measurement and meteorological conditions to O3 concentration? 3. I don't understand the variation of CO concentration during different stages. The authors mentioned fuel combustions should be the reason. Is there any evidence? Why did fuel combustion increase during G20? 4. Other minor errors: Line 61-62: no need to mention "which are dominant compounds of fine particulate matter". Delete it Line 69-70 the complexity of mitigating secondary photochemical pollution is also highly related with intricately photochemical reactions. Thus add the phase "in addition to intricate photochemical reactions" Line 207-210: This section belongs to the description of emission control measures. Thus suggest moving it in Introduction. Line 429-459 The Conclusion is a bit long. The authors are encouraged to shorten this section.
* * *

---

## Author Comment (AC1) · 8 Feb 2020

Based on the unique case of G20 held in Hangzhou, the authors systematically evaluated the effectiveness of powerful control measures implemented by the Chinese State Council on reducing atmospheric primary (i.e., $NO_x$, $SO_2$, and CO) and secondary pollutants (PAN and $O_3$) after discriminating the effect of meteorological condition during G20. Then, they explored the underlying mechanisms of photochemical pollution including PAN and $O_3$ by using MCM, appointed the source of VOCs by PMF model and further calculated the OFP for these various sources. The observational dataset are valuable, and the manuscript reports the measurement results well. In summary, the topic is very interesting and the manuscript is also of good quality. Thus I strongly recommend it could be published in the ACP after minor revision below.

**Response:** Thanks so much for your positive comments on our manuscript. According to your suggestions, we made the corresponding corrections in the revised manuscript.

1) line25-26 reconstruct this sentence as ": : :during G20 Summit provide us a unique opportunity to address this issue. Surface concentrations of: : :"
Accept

2) line 53 add the phase "matter" after "particulate"
Accept

3) line 89-90 rewrite this sentence
According to your suggestion, we have corrected it as "During these events, the effectiveness of a series of emission control measures on reducing atmospheric primary pollutants, in particular to the particulate matter, has been comprehensively evaluated, but less on photochemical pollution." in the revised manuscript.

4) line 120-121 add some detailed information about the $PM_{2.5}$ measurement.
Accept. We added the statement that "Ambient $PM_{2.5}$ samples were collected using co-located Thermo Scientific (formerly R&P) Model 1405D samplers. PM-Coarse and $PM_{2.5}$ particulate, split by a virtual impactor, each accumulate on the system's exchangeable TEOM filters. By maintaining a flow rate of 1.67 L $min^{-1}$ through the coarse sample flow channel and 3 L $min^{-1}$ through the $PM_{2.5}$ sample channel, and measuring the total mass accumulated on each of the TEOM filters, the device can calculate the mass concentration of both the $PM_{2.5}$ and PM Coarse sample streams in near real-time." in the revised manuscript.

5) line 124-125 the abbreviate phase of "EMC" and "EEC" are not consistent with those below. Revise them.
Accept. We revised this sentence as "To quantify the separate effects of meteorological condition (EMC) and emission control measures (ECC) on observed particulate concentrations,…" in the revised manuscript.

---

## Author Comment (AC2) · 8 Feb 2020

The authors evaluated the effectiveness of pollution control measures implemented during the G20 2016 Summit in Hangzhou, China. Field observation on $NO_x$, $SO_2$, CO, VOCs, $PM_{10}$, $PM_{2.5}$, PAN and $O_3$ were carried out. OBM and PMF model tools were used to analyze the data. It's valuable to publish in this journal. However, the English writing should be improved before publication.

**Response:** Thanks a lot for your positive comments and kind work on our manuscript. According to your suggestion, we made the corrections point by point in the revised manuscript.

Specific comments:
Line 269-270: CO showed a gradual increase (~20.7%), which is not consistent with $SO_2$, $NO_2$, and PM. It seems that CO sources are very different with $NO_x$ and $SO_2$ sources in Hangzhou or pollution controls are not effective on CO reduction. Could the authors give more explanations? I also notice that 48i analyzer is used for the measurement. As we know, zero drift is inevitable for this kind of principle. So, pls provide the quality control measures during the observation.

**Response:** As we know, especially in urban region, atmospheric CO is normally derived from human activities (coal combustion, farming, residual usage, etc.) while vehicle exhaust and coal combustion are typically representative of the sources of $NO_x$ and $SO_2$, respectively. As illustrated below in the Section 3.4 in the manuscript, industrial process with coal combustion and vehicle exhaust were strictly limited throughout the whole G20 period. Thereby, $NO_x$ and $SO_2$ both exhibited significant decreases from BG20 to DG20. In addition, straw combustion was excluded according to the decrease in the number of fire spots in the same time period from BG20 to AG20. On the contrary, to ensure the clean energy used in 2016 G20, local government accelerated the supply of liquid natural gas and liquid petroleum gas (ZPSY, 2016, 2017). The consequent CO was more produced from the incomplete combustion of these fuels during G20 relative to BG20. As you speculated, the emission control measures might be poorly effective on CO reduction, specifically on fuel combustion. Also in our study, ethylene, as a representative tracer of fuel combustion, showed continuous increase from BG20 to DG20, further confirming the ineffectiveness of control measures in this source. Therefore, CO showed a gradual increase which is not consistent with the variation of $NO_x$ and $SO_2$. This phenomenon was also found in another research conducted during G20 in 2016 (Zhao et al., 2017).
Yes, all trace gas analyzers were weekly span and daily zero checked during our measurement. Thus, according to your suggestion we added "It is worth noting that CO showed gradual increases (ca. 20.7%) from BG20 to DG20, which was mainly attributed to the weak control in fuel combustion. Specifically, residential usage and liquid natural gas and petroleum gas around YRD regions during this period might account for such unique pattern of CO. The other two types of fuel combustion including straw combustion and coal combustion were both excluded as discussed in Section 3.4." and "All trace gas analyzers were weekly span and daily zero checked." in the revised manuscript, respectively.

*Reference:*

*Zhao, J. P., Luo, L., Zheng, Y. J., Liu, H. H.: Analysis on air quality characteristics and meteorological conditions in Hangzhou during the G20 summit, Acta Scientiae Circumstantiae, 37(10), 3885-3893, 2017. (In Chinese)*

In Fig. 1, TVOCs is needed to add. It seems that $PM_{10}$ and $PM_{2.5}$ results play no roles on the data analysis in the whole context.

**Response:** Yes, as you suggested we added TVOCs in this figure in the revised manuscript.

In this manuscript, we also discussed the variation of PM from BG20 to AG20 and evaluated the effectiveness of powerful control measures on reducing atmospheric pollutants such as PM, PAN, $O_3$, and the other chemicals ($NO_x$, $SO_2$, and CO). As classified in the Introduction, the effectiveness of a series of emission control measures on reducing atmospheric primary pollutants, in particular to the particulate matter, has been comprehensively evaluated during the events such as Summer Olympic Games (August 2008), the 21[th] Asia-Pacific Economic Cooperation (APEC) conference, and China Victory Day Parade (Victory Parade 2015), but less on photochemical pollution. So we focused on their variation and underlying mechanism of photochemical pollution response to the effectiveness of emission control measures. However, it does not mean that PM is not necessary to be investigated in this study. We also paid much attention to PM in terms of their day-to-day variations and estimating the contribution of meteorological conditions by using the simulated $PM_{2.5}$ by WRF-Chem model.

Fig. S1 is better in the manuscript than in the supplement information.
**Response:** Accept

Fig.5, Similar fuel combustion contributions are found in DG20-II and AG20, which is very different with that in BG20. Why?

**Response:** Similar with the explanation response to the first comment above, we speculated that the increased contribution of fuel combustion from BG20 to DG20 II and to AG20 was attributed to the increased supply of liquid natural gas and liquid petroleum gas with the increasingly strict emission control measures on the other fossil fuels during the acceleration of emission control strategy. Similar phenomenon was also found by Li et al. (2015) in APEC China 2014, with the increased contribution of fuel combustion from 7.05 ppbv before APEC to 12.7 ppbv during APEC and to 31.7 ppbv after APEC to VOCs mixing ratios, although the other sources were effectively reduced.

*Reference:*

*Li, J., Xie, S. D., Zeng, L. M., Li, L. Y., Li, Y. Q., and Wu, R. R.: Characterization of ambient volatile organic compounds and their sources in Beijing, before, during, and after Asia-Pacific Economic Cooperation China 2014, Atmos. Chem. Phys., 15, 7945-7959, 2015.*

Much more contents are done in section 3.4 (VOCs source identification and OFP quantification). How do those results relate with the inconsistent variations in the primary and secondary pollutants?

**Response:** In this study, our main objects are not only to discuss the variation of atmospheric primary and secondary pollutants from BG20 to AG20, but especially to elucidate the underlying mechanism for photochemical pollution. We first found the daily maximum average-8 h (DMA8) $O_3$ exhibited a slight increase from BG20 to DG20 I and then decrease from DG20 I to DG20 II and to AG20, which was unlike with the other pollutants. However, we found the peak values of mean daily $O_3$ in DG20 II exhibited significant decrease compared to BG20 and DG20 I. So, another question is proposed, which factors dominated such variation? As we know, VOCs are the crucial precursors of PAN and $O_3$, and thus we should first identify which VOCs were the predominant precursors for PAN and $O_3$ and explore their variation from BG20 to AG20. As depicted in the Introduction, the additional emission control measure was vehicles control. It possibly played an important role in reducing the peak of atmospheric $O_3$ pollution in Hangzhou. Further, we should comprehensively appoint the corresponding sources of various VOCs and compare their variations and their respective ozone formation potentials (OFPs) before, during, and after G20. In summary, VOCs source identification and OFP quantification were necessary for exploring the variation of photochemical pollution from BG20 to AG20 in details.

---

## Author Comment (AC3) · 8 Feb 2020

The manuscript describes a comprehensive observational dataset including atmospheric $O_3$, PAN, particulate matter, VOCs, NOx, and other trace gases to evaluate the effectiveness of emission control measures on reducing pollutant concentrations before, during, and after G20. It's very reasonable to demonstrate the effect of meteorological conditions by using WRF-Chem model. Further, an explicit OBM model was used to identify the predominant VOCs precursors and key chemical processes in PAN and $O_3$ formation and to further appoint the corresponding VOCs sources before, during, and after G20 by using PMF model. The manuscript is clearly written and for matted very well. Thus, after considering several comments below as minor revisions, I recommend the publication of this manuscript in ACP.

**Response:** Thanks so much for your positive comments and kind work on our manuscript. As you suggested, we made the corrections point by point in the revised manuscript.

1. The authors mentioned emission control measures contributed 63.5%, 44.1% and 31.2% to the reductions of $PM_{2.5}$, $SO_2$ and $NO_2$ in DG20 II relative to BG20. And meteorological conditions made negative contributions. What are the other factors contributing to the reduction of the observed pollutants?

**Response:** Your question is quite important. Normally, the pollutant concentration is determined by the strength of emission source, chemical processes, and physical processes (meteorological conditions). In our study, we assumed that no significant change occurred in the chemical processes affecting the concentrations of these primary pollutants from BG20 to AG20. To some extent, the key factors affecting the photochemical reactions such as the intensity of solar irradiation could be indirectly reflected by the meteorological condition. Indeed, we assumed no significant change in the other reactive gases involved in the chemical reactions with these pollutants from BG20 to AG20. Therefore, the variation of the observed pollutants could be roughly attributed to the net contribution of emission control measures and meteorological conditions. In the revised manuscript, we have added "Here we assumed no significant change in chemical processes (specifically the other reactive gases involved in the chemical reactions with these pollutants) from BG20 to AG20." before the estimation.

2. What are the contribution of emission control measure and meteorological conditions to $O_3$ concentration?

**Response:** According to the calculation method as depicted in the manuscript, the contribution of meteorological conditions to the increased $O_3$ concentration was estimated to be 16.4% in this study. For the contribution of emission control measures, it was quite complex and should be separate discussed in different periods. During the period from BG20 to DG20I, the control measures on reducing the emission of VOCs sources except fuel combustion were really effective in alleviating $O_3$ pollution, which is confirmed by the decreased OFP. Unfortunately, during this period the unfavorable meteorological conditions such as the enhanced intensity of solar

irradiation and regional transport both aggravated the $O_3$ pollution. In DG20 II, significant reduction of $NO_x$ due to the additional vehicle controls might lead to the increase in $O_3$ concentration during G20. It was not only because this region was under the VOC-limited regime in Hangzhou revealed by the results of OBM, but also due to the decreased titration effect of NO on $O_3$ in the morning and evening traffic rush hour during this period. These effects significantly worsen the effectiveness of control measures in vehicle exhaust on reducing OFP. Thus, the final contribution of emission control measures to the increased $O_3$ concentration was estimated to be 21.5% in this study.

3. I don't understand the variation of CO concentration during different stages. The authors mentioned fuel combustions should be the reason. Is there any evidence? Why did fuel combustion increase during G20?

**Response:** As we know, atmospheric CO is normally derived from human activities including fuel combustion (coal combustion, farming, residual usage, etc.). As illustrated in the Section 3.4 in the manuscript, industrial process with coal combustion and vehicle exhaust were strictly limited throughout the whole G20 period. In addition, straw combustion was excluded according to the decrease in the number of fire spots in the same time period from BG20 to AG20. On the contrary, to ensure the clean energy used in 2016 G20, local government accelerated the supply of liquid natural gas and liquid petroleum gas (ZPSY, 2016, 2017). The consequent CO was more produced from the incomplete combustion of these fuels during G20 relative to BG20. The emission control measures might be poorly effective on CO reduction, specifically on fuel combustion. Also in our study, ethylene, as a representative tracer of fuel combustion, showed continuous increase from BG20 to DG20, further confirming the ineffectiveness of control measures in this source. Therefore, CO showed a gradual increase. This phenomenon was also found in another research conducted during G20 in 2016 (Zhao et al., 2017).

*Reference:*
*Zhao, J. P., Luo, L., Zheng, Y. J., Liu, H. H.: Analysis on air quality characteristics and meteorological conditions in Hangzhou during the G20 summit, Acta Scientiae Circumstantiae, 37(10), 3885-3893, 2017. (In Chinese)*

4. Other minor errors:
Line 61-62: no need to mention "which are dominant compounds of fine particulate matter". Delete it
Accept

Line 69-70 the complexity of mitigating secondary photochemical pollution is also highly related with intricately photochemical reactions. Thus add the phase "in addition to intricate photochemical reactions".
Accept

Line 207-210: This section belongs to the description of emission control measures.

Thus suggest moving it in Introduction.
Accept.

Line 429-459 The Conclusion is a bit long. The authors are encouraged to shorten this Section.

**Response:** Accept. According to your suggestion, we shorten the Conclusion as "In this study, ground-based concentrations of atmospheric trace gases and particulate matter, together with meteorological parameters, were measured at a NRCS site in urban Hangzhou before, during, and after G20. We found significant decreases in atmospheric VOCs, $PM_{2.5}$, $NO_x$, and $SO_2$ in DG20 relative to BG20 and AG20, respectively, under the unfavorable meteorological conditions (e.g., stable weather pattern and regional transport). This evidence well indicated that the powerful control measures have taken effect in their emissions in Hangzhou. On the contrary, observed DMA8 $O_3$ increased from BG20 to DG20 I, which was attributed to the regional transport from the northern provinces and the enhanced solar radiation intensity, and then decreased from DG20 II to AG20. The decreases in the peak concentration of daily $O_3$ and the OFP estimated from various VOCs sources both suggested the effectiveness of stringent control measures on reducing atmospheric $O_3$ concentrations. Unlike $O_3$, PAN exhibited gradual decrease from BG20 to DG20. With the OBM model, we found acetaldehyde and methyl glyoxal (MGLY) to be the most important second-generation precursors of PAN, accounting for 37.3-51.6% and 22.8%-29.5% of the total production rates. Furthermore, we confirmed that the production of PAN was sensitive to anthropogenic and biogenic VOCs (isoprene) throughout the whole period, specifically aromatics in BG20 and DG20 I but alkenes in AG20. Similarly, the sensitivity of ozone formation was also under VOC-limited regime throughout G20 period. These findings suggest that reducing emissions of alkanes, alkenes, and aromatics would mitigate photochemical smog including PAN and $O_3$ formation. Furthermore, traffic (vehicle exhaust and gasoline evaporation) and industrial sources (solvent utilization, industrial manufacturing, and chemical feedstock) were found to be the major VOCs sources before G20, accounting for ca. 50.0% and 31.7% of the total, respectively, with the ozone formation potential (OFP) of 14.4 ppbv and 16.1 ppbv. Large decreases were found in the sources and OFPs of solvent utilization (74.1% and 17.3%), followed by vehicle exhaust (57.4% and 77.2%) and industrial manufacturing (56.0% and 40.3%) response to the stringent control measures during G20. We also appeal to pay attention on controlling fuel

combustion and biogenic emission especially when anthropogenic VOCs were substantially reduced following the process of control measures." in the revised manuscript.

---

## Author Comment (AC4) · 8 Feb 2020

The revised figures are shown below.

[Figure]

**Fig. 1.** The topography of National Reference Climatological Station (NRCS) (30.22 oN, 120.17 oE, 41.7 m a.s.l) in Hangzhou, China. The pentagram represents the location of NRCS.

**Fig. 2.** The comparisons of daytime PM2.5, NOx, SO2, CO, TVOCs, PAN, DMA8 O3, and DMA8 Ox. before, during, and after G20, denoted as BG20, DG20, and AG20, respectively. The error bars represent the standard de

---

## Referee Report (RR1)

**Referee Report**

All the suggested comments have been carefully addressed in the revised manuscript, and the quality of revised version is correspondingly improved. The revised manuscript is well written in accurate English using the correct synonyms to convey what the authors mean. Therefore, I recommend that this manuscript could be accepted in ACP.